# MMA Regularization: Decorrelating Weights of Neural Networks by Maximizing the Minimal Angles

**Zhennan Wang[†], Canqun Xiang[†], Wenbin Zou[†*], Chen Xu[‡]**
Shenzhen University
{wangzhennan2017, xiangcanqun2018}@email.szu.edu.cn, {wzou, xuchen_szu}@szu.edu.cn

## Abstract

The strong correlation between neurons or filters can significantly weaken the generalization ability of neural networks. Inspired by the well-known Tammes problem, we propose a novel diversity regularization method to address this issue, which makes the normalized weight vectors of neurons or filters distributed on a hypersphere as uniformly as possible, through *maximizing* the *minimal* pairwise *angles* (MMA). This method can easily exert its effect by plugging the MMA regularization term into the loss function with negligible computational overhead. The MMA regularization is simple, efficient, and effective. Therefore, it can be used as a basic regularization method in neural network training. Extensive experiments demonstrate that MMA regularization is able to enhance the generalization ability of various modern models and achieves considerable performance improvements on CIFAR100 and TinyImageNet datasets. In addition, experiments on face verification show that MMA regularization is also effective for feature learning. Code is available at: `https://github.com/wznpub/MMA_Regularization`.

## 1 Introduction

Although neural networks have achieved state-of-the-art results in a variety of tasks, they contain redundant neurons or filters due to the over-parametrization issue [41, 21], which is prevalent in networks [39]. The redundance can lead to catching limited directions in feature space and poor generalization performance [27].

To address the redundancy problem and make neurons more discriminative, some methods are developed to encourage the angular diversity between pairwise weight vectors of neurons or filters in a layer, which can be categorized into the following three types. The first type reduces the redundancy by dropping some weights and then retraining them iteratively during optimization [35, 12, 36], which suffers from complex training scheme and very long training phase. The second type is the widely used orthogonal regularization [38, 52, 23, 51], which exploits a regularization term in loss function to enforce the pairwise weight vectors as orthogonal as possible. However, it has been proven that orthogonal regularization tends to group neurons closer, especially when the number of neurons is greater than the dimension [24], and therefore it only produces marginal improvements [35]. The third type also utilizes a regularization term but to encourage the weight vectors uniformly spaced through minimizing the hyperspherical potential energy [24, 22] inspired from the Thomson problem [47, 44]. Nonetheless, its disadvantage is that both the time complexity and the space complexity are very

---

[†]The authors are with Shenzhen Key Laboratory of Advanced Machine Learning and Applications, Guangdong Key Laboratory of Intelligent Information Processing, Institute of Artificial Intelligence and Advanced Communication, College of Electronics and Information Engineering, Shenzhen University.

[‡]The author is with the Institute of Artificial Intelligence and Advanced Communication, College of Mathematics and Statistics, Shenzhen University.

[*]Corresponding author: Wenbin Zou.

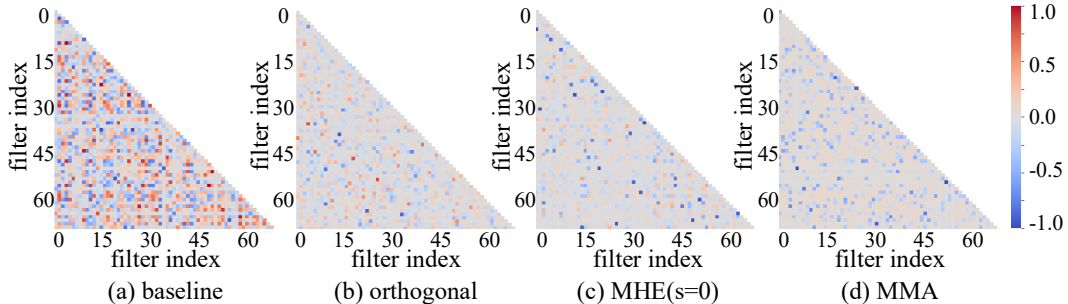

Figure 1: Comparison of filter cosine similarity from the first layer of VGG19-BN trained on CIFAR100 with several different methods of angular diversity regularization. The number of similarity values above 0.2 is 495 (baseline), 120 (orthogonal), 51 (MHE), 0 (MMA), demonstrating the effectiveness of MMA regularization.

high [24], and it suffers from a huge number of local minima and stationary points due to its highly non-convex and non-linear objective function [22].

In this paper, we propose a simple, efficient, and effective method of angular diversity regularization which penalizes the minimum angles between pairwise weight vectors in each layer. Similar to the intuition of the third type mentioned above, the most diverse state is that the normalized weight vectors are distributed on a hypersphere uniformly. To model the criterion of uniformity, we employ the well-known Tammes problem, that is, to find the arrangement of $n$ points on a unit sphere which maximizes the minimum distance between any two points [46, 29, 33, 26, 32]. However, the optimal solutions for the Tammes problem only exist for some combinations of the number of points $n$ and dimensions $d$, which are collected on the N.J.A. Sloane's homepage [43], and obtaining a uniform distribution for an arbitrary combination of $n$ and $d$ is still an open mathematical problem [29]. In this paper, we propose a numerical optimization method to get approximate solutions for the Tammes problem through *maximizing* the *minimal* pairwise *angles* between weight vectors, named as MMA for abbreviation. We further develop the MMA regularization for neural networks to promote the angular diversity of weight vectors in each layer and thus improve the generalization performance.

There are several advantages of MMA regularization: (a) As analyzed in Section 3.2, the gradient of MMA loss is stable and consistent, therefore it is easy to optimize and get near optimal solutions for the Tammes problem as shown in Table 1; (b) As verified in Table 3, the MMA regularization is easy to implement with negligible computational overhead, but with considerable performance improvements; (c) The MMA regularization is effective for both the hidden layers and the output layer, decorrelating the filters and enlarging the inter-class separability respectively. Therefore, it can be applied to multiple tasks, such as image classification and face verification demonstrated in this paper. To intuitively make sense of the effectiveness of MMA regularization, we visualize the cosine similarity of filters from the first layer of VGG19-BN trained on CIFAR100 in Figure 1. We compare several different methods of angular diversity regularization, including orthogonal regularization in [38], MHE regularization in [24], and the proposed MMA regularization. The results show that the MMA regularization gets the most uncorrelated filters. Besides, the MMA regularization keeps some negative correlations which have been verified to be beneficial for neural networks [5].

In summary, the main contributions of this paper are three-fold:

- We propose a numerical method for the Tammes problem, called MMA, which can get near optimal solutions under arbitrary combinations of the number of points and dimensions.

- We develop the novel MMA regularization which effectively promotes the angular diversity of weight vectors and therefore improves the generalization power of neural networks.

- Various experiments on multiple tasks show that MMA regularization is generally effective and can become a basic regularization method for training neural networks.

## 2   Related Work

To improve the generalization power of neural networks, many regularization methods have been proposed to reduce overfitting of neural networks explicitly, such as weight decay [18], decoupled weight decay [25], weight elimination [50], nuclear norm [37], dropout [45], dropconnect [48],

adding noise [2], data augmentation [20], and early stopping [28]. Some new data augmentation methods, such as Cutout [9], RandomErasing [55], and Autoaugment [7], can effectively reduce overfitting and are widely used in many tasks.

The correlation of neurons or filters is prevalent in networks [39]. [16] shows that the correlation of trained filters can be exploited by approximating a learnt full rank filter bank as combinations of rank-1 filter basis, therefore the evaluation procedure can be accelerated. [11] shows that the correlation of filters can be exploited by approximating regular convolutions with depthwise separable convolutions in closed form, therefore the paper proposes network decoupling to accelerate convolutional neural networks by transferring pre-trained models into depthwise separable convolution structure, with a promising speedup yet negligible accuracy loss. [11] introduces the BSConv based on intra-kernel correlations, which allows for a more efficient separation of regular convolutions.

Besides exploiting the correlation, some regularization approaches target at decreasing the inter-kernel correlation. These methods mainly penalize the neural networks by adding a regularization term to the loss function. The regularization term either promotes the diversity of activations through minimizing the cross-covariance of hidden activations [6], or directly promotes the diversity of neurons or filters through enforcing the pairwise orthogonality [38, 52, 23, 51] or minimizing the global potential energy [24, 22]. For many tasks, these methods obtain marginal improvements [38, 35, 53, 4]. Another stream of approaches gets comparatively diverse neurons or filters by cyclically dropping and relearning some of the weights [35, 12, 36], which leads to substantial performance gains, but suffers from complex training. In contrast, our proposed simple MMA regularization achieves significant performance improvements while employing the standard training procedures.

The most related work to our method is MHE [24], which targets the uniform distribution of normalized weight vectors on a hypersphere as well. However, the MHE is inspired by the Thomson problem [47] and models the criterion of uniformity as the minimum global potential energy, which suffers from high computational complexity and lots of local minima [22]. Inspired by the Tammes problem [46, 26], our proposed MMA regularization models the criterion as maximizing the minimum angles, that is the key reason why our method is more efficient and effective.

## 3   MMA Regularization

As our proposed regularization is inspired by the Tammes problem, we firstly analyze the Tammes problem and propose a numerical method called MMA which *maximizes* the *minimal* pairwise *angles* between the vectors. Then we make a comparison of several numerical methods for the Tammes problem by gradient analysis, which demonstrates the advantage of the proposed MMA. Finally, we develop a novel angular diversity regularization for neural networks by the proposed MMA.

### 3.1   The Tammes Problem and Proposed Numerical Method MMA

Construction of points spaced uniformly on a unit hypersphere $S^d \in R^d (d \in \{3,4,5,...\})$ is an important problem for various applications ranging from coding theory to computational geometry [33]. There are many ways to model the criterion of uniformity. One approach is to maximize the minimal pairwise distance between the points [33], i.e.

$$\max \min_{i,j,i \neq j} \|\hat{\mathbf{w}}_i - \hat{\mathbf{w}}_j\|, \qquad s.t. \forall_i \quad \hat{\mathbf{w}}_i = \frac{\mathbf{w}_i}{\|\mathbf{w}_i\|} \tag{1}$$

where $\mathbf{w}_i \in R^{d \times 1}$ denotes the coordinate vector of the $i$-th point, $\hat{\mathbf{w}}$ denotes the $l_2$-normalized vector, and the $\| * \|$ denotes the Euclidean norm. This criterion means the points on a unit sphere are spaced uniformly when the minimal pairwise distance is maximized, which is known as the Tammes problem [46, 26] or the optimal spherical code [10, 43]. Denoting the dimension with $d$ and the number of points with $n$, we firstly analyze the analytical solutions for the case of $d \geq n-1$, and then propose the numerical solutions for the case of $d < n-1$.

**The analytical solutions for $d \geq n-1$.** As the distance between any two points on a unit hypersphere is inversely proportional to the cosine similarity, the Tammes problem is equivalent to minimize the maximal pairwise cosine similarity, i.e.

$$\min \max_{i,j,i \neq j} \hat{\mathbf{w}}_i \cdot \hat{\mathbf{w}}_j, \quad s.t. \forall_i \quad \hat{\mathbf{w}}_i = \frac{\mathbf{w}_i}{\|\mathbf{w}_i\|} \tag{2}$$

The maximum of $\hat{\mathbf{w}}_i \cdot \hat{\mathbf{w}}_j$ must be larger than the average. Therefore, the minimum is derived as:

$$n(n-1) \max_{i,j,i \neq j} \hat{\mathbf{w}}_i \cdot \hat{\mathbf{w}}_j \geq \sum_{i,j,i \neq j} \hat{\mathbf{w}}_i \cdot \hat{\mathbf{w}}_j = \|\sum_i \hat{\mathbf{w}}_i\|^2 - \sum_i \|\hat{\mathbf{w}}_i\|^2 = \|\sum_i \hat{\mathbf{w}}_i\|^2 - n \geq -n \tag{3}$$

Therefore, the minimum of maximal pairwise cosine similarity is $-\frac{1}{n-1}$, which can be reached when all pairwise angles between the points are equal to each other, and the sum of all vectors is a zero vector. This criterion has a matrix form:

$$\boldsymbol{C} = \boldsymbol{\hat{W}}\,\boldsymbol{\hat{W}}^T = \begin{bmatrix} 1 & -\frac{1}{n-1} & \cdots & -\frac{1}{n-1} \\ -\frac{1}{n-1} & 1 & \ddots & \vdots \\ \vdots & \ddots & \ddots & -\frac{1}{n-1} \\ -\frac{1}{n-1} & \cdots & -\frac{1}{n-1} & 1 \end{bmatrix}, \quad s.t. \; \forall_i \quad \boldsymbol{\hat{W}}_i = \frac{\mathbf{w}_i^T}{\|\mathbf{w}_i\|} \tag{4}$$

where $\boldsymbol{\hat{W}} \in R^{n \times d}$ denotes the set of $l_2$-normalized points. According to the matrix theory, the eigenvalues of matrix $\boldsymbol{C}$ are $\lambda_1 = 0$ with algebraic multiplicity of 1 and $\lambda_2 = \frac{n}{n-1}$ with algebraic multiplicity of $n-1$. As all the eigenvalues of $\boldsymbol{C}$ are greater than or equal to zero, $\boldsymbol{C}$ is a semi-positive definite matrix. According to the spectral theorem [3], $\boldsymbol{\hat{W}}$ can be gotten through the eigendecomposition of $\boldsymbol{C}$, which is the analytical solution for the Tammes problem. However, since the rank of $\boldsymbol{C}$ is $n-1$, the rank of $\boldsymbol{\hat{W}}$ and the minimum dimension of the points are also $n-1$. Therefore, this analytical solution only exists for the case of $d \geq n-1$.

**The numerical solutions for $d < n-1$.** So far, under the case of $d < n-1$, the analytical solutions for the Tammes problem only exist for some combinations of $n$ and $d$ [43]. For most combinations, the optimal solutions do not exist. Consequently, numerical methods are used to get approximate solutions.

As the objective (Equation 1) of Tammes problem is not globally differentiable [34], the conventional solution [1] alternatively optimizes a differentiable potential energy function to get the approximate solutions, as discussed in next subsection. Nonetheless, with the help of SGD [40] and modern automatic differentiation library [31], we can now directly use Equation (1) to implement optimization and get the approximate solutions. However, the calculation of Euclidean length is expensive. Alternatively, as mentioned in Equation (2), we can use the cosine similarity as the objective function, called cosine loss, which is formulated as follows:

$$l_{cosine} = \frac{1}{n} \sum_{i=1}^{n} \max_{j, j \neq i} \boldsymbol{Cos}_{ij}, \quad \boldsymbol{Cos} = \boldsymbol{\hat{W}}\,\boldsymbol{\hat{W}}^T, \quad s.t. \; \forall_i \quad \boldsymbol{\hat{W}}_i = \frac{\mathbf{w}_i^T}{\|\mathbf{w}_i\|} \tag{5}$$

where $\boldsymbol{Cos} \in R^{n \times n}$ denotes the cosine similarity matrix of the points. Employing the global maximum similarity as Equation (2) is inefficient, as it only updates the closest pair of points. Therefore, we alternatively use the average of each vector's maximum similarity.

The cosine loss can be optimized quickly taking the advantage of matrix form. However, we find this loss is hard to converge, especially for the case that $\mathbf{w}_i$ is very close to $\mathbf{w}_j$, which is very prevalent in neural networks [39]. As analyzed in next subsection, this is because the gradient is too small to cover random fluctuations during the optimization. Gaining insight from the ArcFace [8], we propose the angular version of cosine loss as the object function:

$$l_{MMA} = -\frac{1}{n} \sum_{i=1}^{n} \min_{j, j \neq i} \boldsymbol{\theta}_{ij}, \quad \boldsymbol{\theta} = \arccos(\boldsymbol{\hat{W}}\,\boldsymbol{\hat{W}}^T), \quad s.t. \; \forall_i \quad \boldsymbol{\hat{W}}_i = \frac{\mathbf{w}_i^T}{\|\mathbf{w}_i\|} \tag{6}$$

where $\boldsymbol{\theta} \in R^{n \times n}$ denotes the pairwise angle matrix. As this loss *maximizes* the *minimal* pairwise *angles*, we name it MMA loss for abbreviation. The MMA loss is very efficient and robust for optimization, so it is easy to get near optimal numerical solutions for the Tammes problem. Besides, it can also get close solutions for the case $d \geq n-1$, which is validated in Section 4. In next subsection, we demonstrate the advantage of the proposed MMA loss through gradient analysis and comparison.

### 3.2 The Gradient Analysis

In this subsection, we analyze and compare the gradients of loss functions generating approximate solutions for uniformly spaced points. To simplify the derivation, we only consider the norm of the gradient of the core function, composing the summation in loss functions, w.r.t. corresponding weight vector $\mathbf{w}_i$. For intuitive comparison, the analysis results are presented in Figure 2.

Corresponding to the cosine loss referred to Equation (5), the gradient norm is derived as follows:

$$\left\| \frac{\partial \boldsymbol{Cos}_{ij}}{\partial \mathbf{w}_i} \right\| = \left\| \frac{\partial \left( \frac{\mathbf{w}_i^T \mathbf{w}_j}{\|\mathbf{w}_i\| \|\mathbf{w}_j\|} \right)}{\partial \mathbf{w}_i} \right\| = \frac{\|(\boldsymbol{I} - \boldsymbol{M}_{\mathbf{w}_i})\mathbf{w}_j\|}{\|\mathbf{w}_i\| \|\mathbf{w}_j\|} = \frac{\|\mathbf{w}_j\| \sin \boldsymbol{\theta}_{ij}}{\|\mathbf{w}_i\| \|\mathbf{w}_j\|} = \frac{\sin \boldsymbol{\theta}_{ij}}{\|\mathbf{w}_i\|}, \quad \boldsymbol{M}_{\mathbf{w}_i} = \frac{\mathbf{w}_i \mathbf{w}_i^T}{\|\mathbf{w}_i\|^2} \tag{7}$$

where $M_{\mathbf{w}_i}$ represents the projection matrix of $\mathbf{w}_i$. From the above derivation and Figure 2, we can see that the gradient norm is very small when pairwise angle is close to zero. That is why the cosine loss is hard to converge for the case that $\mathbf{w}_i$ and $\mathbf{w}_j$ are close to each other, as experimented in Section 4. Next, we derive the gradient norm corresponding to the MMA loss referred to Equation (6):

$$\|\frac{\partial \boldsymbol{\theta}_{ij}}{\partial \mathbf{w}_i}\| = \|\frac{\partial \boldsymbol{\theta}_{ij}}{\partial \cos \boldsymbol{\theta}_{ij}} \frac{\partial \cos \boldsymbol{\theta}_{ij}}{\partial \mathbf{w}_i}\| = \frac{1}{\sin \boldsymbol{\theta}_{ij}} \frac{\sin \boldsymbol{\theta}_{ij}}{\|\mathbf{w_i}\|} = \frac{1}{\|\mathbf{w}_i\|} \tag{8}$$

Compared to the gradient norm corresponding to the cosine loss, as referred to Equation (7), the gradient norm corresponding to the MMA loss is independent of the pairwise angle $\theta_{ij}$, so it would not encounter the very small gradient even though $\theta_{ij}$ is zero. Figure 2 shows that the gradient norm corresponding to MMA loss is stable and consistent. Therefore, the MMA loss is easy to optimize and get near optimal solutions for the Tammes problem, verified by experiments in Section 4.

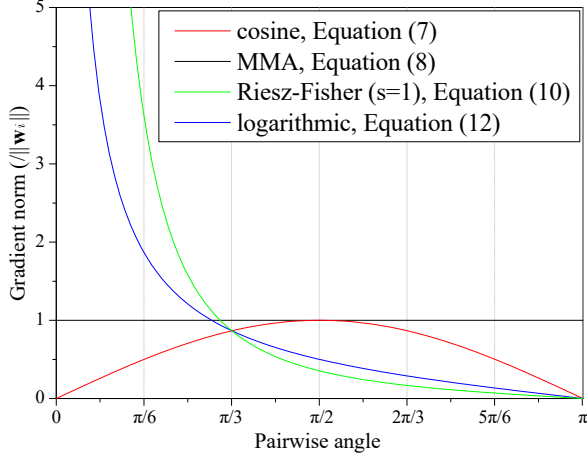

Figure 2: Comparison of the gradient norm changed with pairwise angle. The gradient of MMA loss is stable and consistent.

In addition to the above two loss functions, we also analyze the Riesz-Fisher loss [33] and the logarithmic loss [33] which are often used to get uniformly distributed points on a hypersphere. The philosophy behind the two loss functions is that the points on a hypersphere are uniformly spaced when the potential energy is minimum, and both of them are formulated as kernel functions of the potential energy. The Riesz-Fisher loss and the corresponding gradient norm are:

$$l_{RF} = \frac{1}{n(n-1)} \sum_{i \neq j} \|\hat{\mathbf{w}}_i - \hat{\mathbf{w}}_j\|^{-s}, \ s > 0 \tag{9}$$

$$\|\frac{\partial \|\hat{\mathbf{w}}_i - \hat{\mathbf{w}}_j\|^{-s}}{\partial \mathbf{w}_i}\| = \frac{s}{\|\mathbf{w}_i\|} \frac{\cos \frac{\boldsymbol{\theta}_{ij}}{2}}{(2 \sin \frac{\boldsymbol{\theta}_{ij}}{2})^{s+1}} \tag{10}$$

where $s$ is a hyperparameter, and is set to 1 in Figure 2 for easy comparison. The logarithmic loss and the corresponding gradient norm are:

$$l_{log} = -\frac{1}{n(n-1)} \sum_{i \neq j} \log \|\hat{\mathbf{w}}_i - \hat{\mathbf{w}}_j\| \tag{11}$$

$$\|\frac{\partial \log \|\hat{\mathbf{w}}_i - \hat{\mathbf{w}}_j\|}{\partial \mathbf{w}_i}\| = \frac{1}{\|\mathbf{w}_i\|} \frac{\cos \frac{\boldsymbol{\theta}_{ij}}{2}}{(2 \sin \frac{\boldsymbol{\theta}_{ij}}{2})} \tag{12}$$

Due to the limited space, more details of derivation are presented in the supplementary material. As visualized in Figure 2, the Riesz-Fisher loss and logarithmic loss have similar properties: the gradient norm is sharp around angles near zero and drops rapidly as the angle increases. Besides, the greater the $s$ of Riesz-Fisher loss is, the sharper the gradient norm becomes. The very large gradient norm around angles near zero can cause instability and prevent the normal learning of neural networks, and the very small gradient norm around angles away from zero makes the updates inefficient. We argue that is why the two loss functions just get inaccurate solutions for the Tammes problem in Section 4 and perform not so good in terms of accuracy in Table 3.

## 3.3 MMA Regularization for Neural Networks

In this subsection, we develop the MMA regularization for neural networks, which promotes the learning towards uniformly distributed weight vectors in angular space. For $d \geq n - 1$, we can employ the cosine similarity matrix in Equation (4) to constrain the weights. However, as the MMA loss can generate accurate approximate solutions in any case and is easy to implement, we uniformly exploit the MMA loss referred to Equation (6) as the angular regularization:

$$l_{MMA\_regularization} = \lambda \sum_{i=1}^{L} l_{MMA}(\boldsymbol{W}_i) \tag{13}$$

where $\lambda$ denotes regularization coefficient, $L$ denotes the total number of layers, including convolutional layers and fully connected layers, and $\boldsymbol{W}_i$ denotes the weight matrix of the $i$-th layer with each row denoting a vectorized filter or neuron.

The MMA regularization is complementary and orthogonal to weight decay [18]. Weight decay regularizes the Euclidean norm of weight vectors, while MMA regularization promotes the direction diversity of weight vectors. MMA regularization can be applied into both hidden layers and output layer. For hidden layers, MMA regularization can reduce the redundancy of filters, which is very common in neural networks [39]. Consequently, the unnecessary overlap in the features captured by the network's filters is diminished. For output layer, MMA regularization can maximize the inter-class separability and therefore enhance the discriminative power of neural networks.

## 4   Experiments for the Tammes Problem

This section compares several numerical methods for the Tammes problem, measured by the minimum angle, as shown in Table 1. The first column denotes the dimension $d$ and the second column denotes the number of points $n$. The third column refers the minimal pairwise angles of the optimal solutions collected in [43]. The rest columns are the minimum angle obtained by several different numerical methods, including MMA loss in Equation (6), cosine loss in Equation (5), Riesz-Fisher loss with $s = 2$ in Equation (9), and logarithmic loss in Equation (11). The weights are initialized with values drawn from the standard normal distribution and then optimized by SGD [40] with 10000 iterations. The initial learning rate is set to 0.1 and reduced by a factor of 5 once learning stagnates, and the momentum is set to 0.9.

For $d \geq n - 1$, as analyzed in Section 3.1, each pairwise angle of optimal solutions is $arccos(-\frac{1}{n-1})$, verified by the second row ($d$=3, $n$=4), the fifth row ($d$=4, $n$=5), and the ninth row ($d$=5, $n$=6), from which we can observe that the optimal solutions can be easily achieved by any of the four numerical methods. For $d < n - 1$, all the numerical solutions are more or less prone to be worse than the optimal solutions. However, the MMA loss can robustly obtain the closest solutions to the optimal. The cosine loss can also achieve very close solutions, but it is not robust for the cases of too many

Table 1: Minimum angle (degree) obtained by several different loss functions for the Tammes problem. The best results are highlighted in bold.

| $d$ | $n$ | optimal | $l_{MMA}$ | $l_{cosine}$ | $l_{RF}$ | $l_{log}$ |
|-----|-----|---------|-----------|--------------|----------|-----------|
| 3 | 4 | 109.5 | **109.5** | **109.5** | 109.4 | **109.5** |
| 3 | 30 | 38.6 | **38.5** | 0 | 34.9 | 35.4 |
| 3 | 130 | 18.5 | **17.6** | 0 | 12.6 | 16.7 |
| 4 | 5 | 104.5 | **104.5** | **104.5** | **104.5** | **104.5** |
| 4 | 30 | 54.3 | **54.0** | 53.7 | 49.3 | 48.6 |
| 4 | 130 | 33.4 | 32.0 | **32.1** | 27.9 | 27.2 |
| 4 | 600 | 19.8 | **19.3** | 0 | 15.9 | 13.1 |
| 5 | 6 | 101.5 | **101.5** | **101.5** | 101.2 | **101.5** |
| 5 | 30 | 65.6 | **65.5** | 64.0 | 57.1 | 60.0 |
| 5 | 130 | 43.8 | **42.9** | 42.5 | 35.8 | 35.6 |

points like the third row ($d$=3, $n$=30), the forth row ($d$=3, $n$=130), and the eighth row ($d$=4, $n$=600). This is due to the too small gradient as analyzed in Section 3.2. The Riesz-Fisher loss and the logarithmic loss are also robust, but they converge to solutions far from the optimal.

## 5   Experiments on Image Classification

### 5.1   Implementation Settings

We conduct image classification experiments on CIFAR100 [17] and TinyImageNet [19]. For both datasets, we follow the simple data augmentation in [20]. We employ various classic networks as the backbone networks, including ResNet56 [13], VGG19 [42] with batch normalization [15] denoted by VGG19-BN, VGG16 with batch normalization denoted by VGG16-BN, WideResNet [54] with 16 layers and a widen factor of 8 denoted by WRN-16-8, and DenseNet [14] with 40 layers and a growth rate of 12 denoted by DenseNet-40-12. We denote the corresponding MMA regularization version of models by X-MMA. For fair comparison, not only the X-MMA models but also the corresponding backbones are trained from scratch, so our results may be slightly different from the ones presented in the original papers due to different random seeds and hardware settings.

For CIFAR100, the hyperparameters and settings are the same as the original papers. For example, the batch size is set to 64 for DenseNet and 128 for other models, the learning rate is initially set to 0.1 and decayed by specific schedules, and the optimizer is SGD with a momentum of 0.9. For TinyImageNet, we follow the settings in [49]. Besides, all the random seeds are fixed, so the experiments are

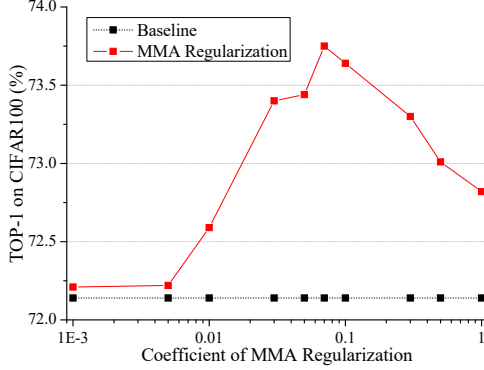
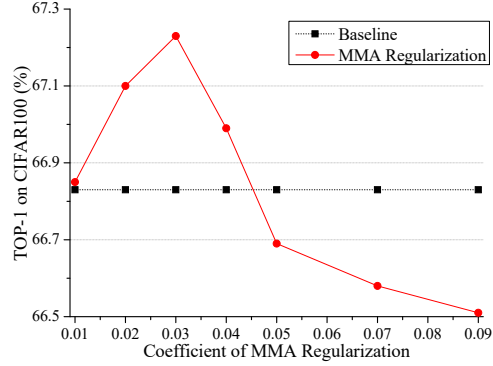

Figure 3: Coefficient tuning for VGG19-BN.

Figure 4: Coefficient tuning for ResNet20.

reproducible and comparisons are absolutely fair. Except otherwise noted, we employ the average accuracy of last five epoches as the evaluation criterion to reduce the variance of evaluation.

## 5.2 Ablation Study

To understand the behavior of MMA regularization, we conduct comprehensive ablation experiments on CIFAR100. Except otherwise noted, we use the VGG19-BN to implement ablation experiments.

**Impact of the hyperparameter.** The MMA regularization coefficient $\lambda$ is the only hyperparameter. As the skip connections have implicitly promoted the angular diversity of neurons [30], we separately select the VGG19-BN and ResNet20 to investigate the impact of different coefficients for models without and with skip connections, as shown in Figure 3 and Figure 4 respectively. To better clarify the influences, we report the mean of final test accuracy at the end of training over 5 runs using random seeds ranging from 123 to 523 with an interval of 100. From both of the figures, we can see that the effect of MMA regularization with too small coefficients is not obvious. However, too large coefficients improve slightly or even decrease the performance. This is because too strong regularization prevents the normal learning of neural networks to some extent. For the VGG19-BN, MMA regularization is not very sensitive to the hyperparameter and works well from 0.03 to 0.2, therefore proving the effectiveness of MMA regularization. For the ResNet20, it is sensitive because of the skip connections. In the following experiments, we set MMA regularization coefficient to 0.07 for VGG models and 0.03 for models with skip connections.

**Effectiveness for hidden layers and output layer.** The MMA regularization is applicable to both the hidden layers and the output layer. In Table 2, we study the effect of MMA regularization applied to hidden layers (*hidden*) and all layers (*hidden+output*). The results show that the *hidden* version improves over the VGG19-BN baseline with a considerable margin and, moreover, the *hidden+output*

Table 2: Accuracy (%) of applying MMA regularization to different layers.

| Model | TOP-1 | TOP-5 |
|---|---|---|
| baseline | 72.08 | 90.5 |
| hidden | 73.45 | 90.91 |
| hidden+output | **73.73** | **91.21** |

version improves the performance further. This indicates that the MMA regularization is effective for both the hidden layers and the output layer, and the effects can be accumulated. As analyzed in Section 3.3, the effectiveness for hidden layers comes from decorrelating the filters or neurons, and the effectiveness for output layer comes from enlarging the inter-class separability.

**Comparison with other angular regularization.** This section compares several angular regularization from the perspective of calculating time per batch, occupied memory, accuracy, and the minimum pairwise angles of several layers, as shown in Table 3. Besides the MMA regularization, we also consider the MHE [24] regularization and the widely used orthogonal regularization [38, 52, 23, 51] which also penalize the pairwise angles. The MHE actually takes the Riesz-Fisher loss (s>0) or logarithmic loss (s=0) to implement regularization [24]. The orthogonal regularization promotes all the pairwise weight vectors to be orthogonal. Here, we adopt the orthogonal regularization in [38]:

$$l_{orthogonal} = \frac{\lambda}{2} \sum_{i=1}^{L} \| \hat{\boldsymbol{W}}_i \hat{\boldsymbol{W}}_i^T - \boldsymbol{I} \|_F^2 \qquad (14)$$

where $\hat{\boldsymbol{W}}_i$ denotes the $l_2$-normalized weight matrix of the $i$-th layer, $\boldsymbol{I}$ denotes identity matrix, and $\| * \|_F$ denotes the Frobenius norm.

Table 3: Comparison of several different methods of angular regularization. The MMA achieves the most diverse filters and highest accuracy with negligible computational overhead.

| Regularization | Time(s)/Batch | Memory (MiB) | Accuracy (%) | | Minimum Angle (degree) | | | |
|---|---|---|---|---|---|---|---|---|
| | | | TOP-1 | TOP-5 | L3-3 | L4-3 | L5-3 | Classify |
| baseline | 0.070 | 1127 | 72.08 | 90.50 | 70.1 | 16.0 | 30.8 | 54.0 |
| MMA | 0.095 | 1229 | **73.73** | **91.21** | 85.7 | 84.7 | 85.3 | 84.9 |
| MHE(s=2) | 0.259 | 5551 | 71.91 | 90.76 | 74.2 | 58.1 | 68.5 | 71.3 |
| MHE(s=0) | 0.253 | 5551 | 72.08 | 90.72 | 69.6 | 48.6 | 57.8 | 63.4 |
| orthogonal | 0.096 | 1237 | 72.83 | 90.98 | 79.3 | 75.3 | 81.2 | 63.1 |

The coefficient is set to 0.07 for MMA, 1.0 for MHE [24], and 0.0001 for orthogonal regularization [51]. Due to the limit of GPU memory, we employ the mini-batch version of MHE [24], which iteratively takes a random batch (here, 30% of the total) of weight vectors to calculate the loss. For the comparison of minimal pairwise angle, we select the third layer of the third block, forth block, and fifth block, and the classification layer, which are denoted by *L3-3*, *L4-3*, *L5-3*, and *Classify* respectively. These experiments are based on PyTorch [31] and NVIDIA GeForce GTX 1080 GPU.

Compared to the baseline, the MMA regularization and orthogonal regularization slightly increase the calculating time and occupied memory. However, the MHE regularization greatly increases that due to the computation of all the pairwise distances. In terms of accuracy, the MMA regularization improves over the baseline by a substantial margin. The orthogonal regularization is also effective but inferior to the MMA regularization. The MHE regularization is just comparable to the baseline, which may be because of the unstable gradient as analyzed in Section 3.2. We also observe that there is a strong link between the minimal pairwise angles in hidden layers and the accuracy—the larger the minimal angles, the higher the accuracy. This is because the larger minimal angle means the more diverse filters which would improve the generalizability of models. The MMA regularization is also the most effective to enlarge the minimal pairwise angle of classification layer, which would increase the inter-class separability and enhance the discriminative power of neural networks. More plots and comparison of the minimal pairwise angles are shown in the supplementary material.

**Combined with other regularization methods.** As we exactly follow the same settings in the original papers proposing the models, weight decay and data augmentation have been applied to all the classification models in this paper, so the results in Table 5 and Table 6 prove that MMA can be combined with weight decay and simple data augmentation. To further demonstrate this advantage, we perform comparative experiments with WRN-28-10 on CIFAR100 in Table 4. We firstly test the Autoaugment [7], the SOTA data augmentation method, then combine the MMA with it. The results show that Autoaugment is effective and MMA can further improve the accuracy. All these demonstrate that MMA can be combined with other regularization methods to further improve the test performance.

Table 4: Combining MMA with Autoaugment based on WRN-28-10 on CIFAR100.

| Model | Accuracy |
|---|---|
| WRN-28-10 | 80.65 |
| WRN-28-10-Autoaugment | 82.65 |
| WRN-28-10-Autoaugment-MMA | **83.11** |

## 5.3 Results and Analysis

We firstly compare various modern architectures with their MMA regularization versions on CIFAR100. From the results shown in Table 5, we can see that the X-MMA can typically improve the corresponding backbone models. Especially, MMA regularization improves the TOP-1 accuracy of VGG19-BN by 1.65%. MMA regularization is also able to robustly improve the performance of models with skip connections like ResNet, DenseNet, and WideResNet, although the improvement is not as distinct as in VGG. This is because the skip connections have implicitly reduced feature correlations to some extent [30].

Table 5: Accuracy (%) on CIFAR100.

| Model | TOP-1 | TOP-5 |
|---|---|---|
| ResNet56 | 70.39 | 91.12 |
| ResNet56-MMA | **70.90** | **91.25** |
| VGG19-BN | 72.08 | 90.50 |
| VGG19-BN-MMA | **73.73** | **91.21** |
| WRN-16-8 | 78.97 | 94.84 |
| WRN-16-8-MMA | **79.34** | **95.05** |
| DenseNet-40-12 | 73.98 | 92.74 |
| DenseNet-40-12-MMA | **74.61** | **92.77** |

To further demonstrate the consistency of MMA's superiority, we also evaluate the MMA regularization with ResNet56 and VGG16-BN on TinyImageNet, with the coefficient of 0.01 and

0.07 respectively. The results are reported in Table 6, where the X-MMA models successfully outperform the original backbones on both Top-1 and Top-5 accuracy. It is worth emphasizing that the X-MMA models achieve the improvements with quite negligible computational overhead and without modifying the original architecture.

Table 6: Accuracy (%) on TinyImageNet.

| Model | TOP-1 | TOP-5 |
|---|---|---|
| ResNet56 | 54.80 | 78.71 |
| ResNet56-MMA | **55.24** | **78.92** |
| VGG16-BN | 62.16 | 82.41 |
| VGG16-BN-MMA | **63.37** | **82.68** |

To intuitively illustrate the effectiveness of the MMA regularization, we plot the training curves of VGG19-BN and ResNet56 on CIFAR100 in Fig. 5 and Fig. 6 respectively. The MMA can get persistently higher TOP-1 accuracy than the baseline. Besides, the convergence speed and stability are not changed.

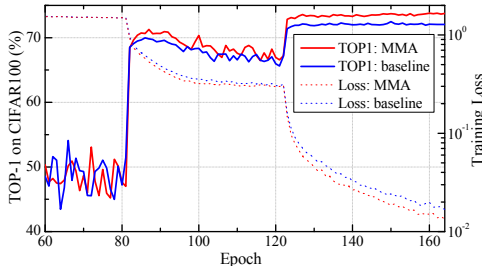

Figure 5: Training curves of VGG19-BN.

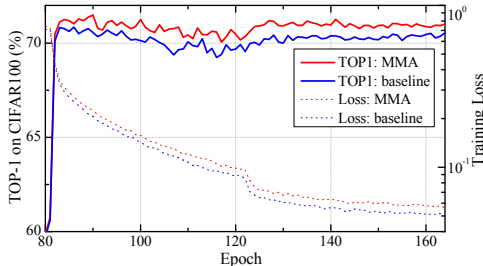

Figure 6: Training curves of ResNet56.

## 6 ArcFace+: Applying MMA Regularization to ArcFace

ArcFace [8] is one of the state-of-the-art face verification methods, which proposes an additive angular margin between the learned feature and the target weight vector in the classification layer. This method essentially encourages intra-class feature compactness by promoting the learned features close to the target weight vectors. As analyzed in Section 3.3, MMA regularization can achieve diverse weight vectors and therefore improve inter-class separability for classification layer. Consequently, the MMA regularization is complementary to the objective of ArcFace and should boost accuracy further. Motivated by this analysis, we propose ArcFace+ by applying MMA Regularization to ArcFace. The objective function of ArcFace+ is defined as:

$$l_{arcface+} = l_{arcface}(m) + \lambda l_{MMA}(\boldsymbol{W}_{classify}) \tag{15}$$

where $m$ is the angular margin of ArcFace, $\lambda$ is the regularization coefficient, and $\boldsymbol{W}_{classify}$ is the weight matrix of classification layer.

For fair comparison, both the ArcFace and ArcFace+ are trained from scratch, therefore our results of the ArcFace may be slightly different from the ones presented in the original paper due to different settings and hardware. The implementation settings are detailed in the supplementary material.

From the results shown in Table 7, we can see that the ArcFace+ outperforms ArcFace across all the three verification datasets by margins which are very significant in the field of face verification. This comparison validates the effectiveness of

Table 7: Comparison of verification results (%).

| Method | LFW | CFP-FP | AgeDB-30 |
|---|---|---|---|
| ArcFace | 99.35 | 95.30 | 94.62 |
| ArcFace+ | **99.45** | **95.59** | **95.15** |

MMA regularization in feature learning. Note that these results are obtained with the default coefficient 0.03, we argue the results may be better with hyperparameter tuning.

## 7 Conclusion

In this paper, we propose a novel regularization method for neural networks, called MMA regularization, to encourage the angularly uniform distribution of weight vectors and therefore decorrelate the filters or neurons. The MMA regularization has stable and consistent gradient, and is easy to implement with negligible computational overhead, and is effective for both the hidden layers and the output layer. Extensive experiments on image classification demonstrate that the MMA regularization is able to enhance the generalization power of neural networks by considerable improvements. Moreover, MMA regularization is also effective for feature learning with significant margins, due to enlarging the inter-class separability. As the MMA can be viewed as a basic regularization method for neural networks, we will explore the effectiveness of MMA regularization on other tasks, such as object detection, object tracking, and image captioning, etc.

## Broader Impact

This work does not present any foreseeable societal consequence.

## Acknowledgments and Disclosure of Funding

This work was supported in part by the NSFC Project (61872429 and 61771321), in part by the key Project of DEGP (2018KCXTD027), in part by the Natural Science Foundation of Guangdong Province (2020A1515010959), in part by Natural Science Foundation of Shenzhen (JCYJ20170818091621856 and JCYJ2020N294) and in part by the Interdisciplinary Innovation Team of Shenzhen University.

Besides, we are grateful to Yuanman Li and Zhengyu Zhang for their suggestions.

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
