[Supplementary Material · supplementary_material.pdf]

# Supplementary Material

## A   Detail Derivation of Equation (10) and Equation (12)

The detail derivation of Equation (10) is as follows:

$$\|\frac{\partial \|\hat{\mathbf{w}}_i - \hat{\mathbf{w}}_j\|^{-s}}{\partial \mathbf{w}_i}\|$$

$$=\|\frac{\partial \|\hat{\mathbf{w}}_i - \hat{\mathbf{w}}_j\|^{-s}}{\partial \|\hat{\mathbf{w}}_i - \hat{\mathbf{w}}_j\|} \frac{\partial \|\hat{\mathbf{w}}_i - \hat{\mathbf{w}}_j\|}{\partial (\hat{\mathbf{w}}_i - \hat{\mathbf{w}}_j)} \frac{\partial (\hat{\mathbf{w}}_i - \hat{\mathbf{w}}_j)}{\partial \hat{\mathbf{w}}_i} \frac{\partial \frac{\mathbf{w}_i}{\|\mathbf{w}_i\|}}{\partial \mathbf{w}_i}\|$$

$$=\|\frac{s}{\|\hat{\mathbf{w}}_i - \hat{\mathbf{w}}_j\|^{s+1}} \frac{(\hat{\mathbf{w}}_i - \hat{\mathbf{w}}_j)^T}{\|\hat{\mathbf{w}}_i - \hat{\mathbf{w}}_j\|} I \frac{(I - M_{\mathbf{w}_i})}{\|\mathbf{w}_i\|}\|$$

$$=\frac{\|s(I - M_{\mathbf{w}_i})\hat{\mathbf{w}}_j\|}{\|\mathbf{w}_i\|\|\hat{\mathbf{w}}_i - \hat{\mathbf{w}}_j\|^{s+2}}$$

$$=\frac{s}{\|\mathbf{w}_i\|} \frac{\sin\theta_{ij}}{(2\sin\frac{\theta_{ij}}{2})^{s+2}}$$

$$=\frac{s}{\|\mathbf{w}_i\|} \frac{\cos\frac{\boldsymbol{\theta}_{ij}}{2}}{(2\sin\frac{\boldsymbol{\theta}_{ij}}{2})^{s+1}}, \qquad with \quad M_{\mathbf{w}_i} = \frac{\mathbf{w}_i\mathbf{w}_i^T}{\|\mathbf{w}_i\|^2}$$

The detail derivation of Equation (12) is as follows:

$$\|\frac{\partial \log\|\hat{\mathbf{w}}_i - \hat{\mathbf{w}}_j\|}{\partial \mathbf{w}_i}\|$$

$$=\frac{1}{\|\hat{\mathbf{w}}_i - \hat{\mathbf{w}}_j\|}\|\frac{\partial \|\hat{\mathbf{w}}_i - \hat{\mathbf{w}}_j\|^{-(-1)}}{\partial \mathbf{w}_i}\|$$

$$=\frac{1}{\|\hat{\mathbf{w}}_i - \hat{\mathbf{w}}_j\|} \frac{|-1|}{\|\mathbf{w}_i\|} \frac{\cos\frac{\boldsymbol{\theta}_{ij}}{2}}{(2\sin\frac{\boldsymbol{\theta}_{ij}}{2})^{-1+1}}$$

$$=\frac{1}{\|\mathbf{w}_i\|} \frac{\cos\frac{\boldsymbol{\theta}_{ij}}{2}}{(2\sin\frac{\boldsymbol{\theta}_{ij}}{2})}$$

## B   Dataset Description of Section 5

We conduct our image classification experiments on CIFAR100 [4] and TinyImageNet [5]. The CIFAR100 consists of 50k and 10k images of $32 \times 32$ pixels for the training and test sets respectively. We present experiments trained on the training set and evaluated on the test set. The TinyImageNet dataset is a subset of the ILSVRC2012 classification dataset [8]. It consists of 200 object classes, and each class has 500 training images, 50 validation images, and 50 test images. All images have been downsampled to $64 \times 64$ pixels. As the labels for test set are not released, we present experiments trained on the training set and evaluated on the validation set. For both datasets, we follow the simple data augmentation in [6]. For training, 4 pixels are padded on each side and a $32 \times 32$ crop for CIFAR100 or a $64 \times 64$ crop for TinyImageNet is randomly sampled from the padded image or its horizontal flip. For testing, we only evaluate the single view of the original $32 \times 32$ image for CIFAR100 or $64 \times 64$ image for TinyImageNet. Note that our focus is on the effectiveness of our proposed MMA regularization, not on pushing the state-of-the-art results, so we do not use any more data augmentation and training tricks to improve accuracy.

## C   Implementation Settings of Section 6

We employ CASIA [10] as training dataset and LFW [3], CFP-FP [9], and AgeDB-30 [7] as face verification datasets. For the embedding network, we employ ResNet50 [2]. The angular margin $m$ is set to 0.5 according to the ArcFace paper [1]. The regularization coefficient $\lambda$ is set to 0.03. Other hyperparameters and settings exactly follow the ArcFace paper [1], except for the batchsize and learning schedule. Due to the limit of hardware, we set the batch size to 440 (the ArcFace paper sets to 512). Accordingly, we finish the training process at 38K iterations and decay the learning rate by a factor of 10 at 23750 and 33250 iterations to ensure the same training samples.

# D  Supplement to Section 5.2: Comparison of the Minimal Pairwise Angle

Figure 1: Comparison of the minimal pairwise angle from all layers of VGG19-BN trained on CIFAR100 with several different diversity regularization. The MMA regularization gets the largest minimal pairwise angle consistently across all layers, and therefore the most diverse weight vectors.