[Reviews · NeurIPS 2020]

Review 1

Summary and Contributions: For naively trained neural networks, the neurons at each layer could be strongly correlated, i.e. the input weight vectors align with each other very well. This paper claimed that this "strong correlation" could hurt the test performance. To address this issue, the authors proposed to maximize the minimal pairwise angle (MMA) of input weight vectors for each hidden layer.

Strengths: The experiments shows consistent improvements over the previous diveristy-enhancing regularizations. In addition, the extra computation overhead of the MMA regularization is also negligible. It seems promising to apply this regularization to other problems.

Weaknesses: The contribution of this paper seems incremental. The neuron diversity issue was already raised and investigated in previous studys. Moreover, as far as I can tell, the improvement of test performances over the other methods is not that significant. Please correct me if I misunderstood it. In addition, it is not clear if the new regularization can be combined with other popular regularization methods, such as weight decay, data argumentation, etc. The author may need to compare with SOTA or consider other datasets, for which the improvement becomes either significant, or MMA can be combined with other regularization technique to further improve the test performance.

Correctness: The claim that the lack of neuron diveristy can hurt the generalization performance is indeed a good motivation to propose the new method. However, it is not supported by any theoretical or numerical evidences. It is really doubtful that the improvement of accuracy actually come from the increasing of neuron diversity. Afterall, the new regularization might introduce some other effects we don't know. More ablation studies are needed to identify the real reason for the improvement.

Clarity: the paper is well written and organized. I can easily follow the author's idea.

Relation to Prior Work: Yes

Reproducibility: No

Additional Feedback: The authors may need to provide the details of the training procedure for experiments in Section 5. It is well-known that the batch size, learning rate and the optimizer could significantly affects the test performance for these problems. == After rebutal == The authors' response addressed my concern about the significance of improvements. I raised my score to 6 considering that this regularizer can easily and efficiently applied to different problems.


Review 2

Summary and Contributions: This paper proposes a numerical method for the Tammes problem. By maximizing the minimal pairwise angles, the method can achieve near-optimal solutions under different settings. The method can then be easily applied in deep neural networks as an extra regularization loss to promote angular diversity between pairwise weight vectors. Experimental results show that the proposed method can improve the performance in both image classification and face recognition tasks.

Strengths: 1. This paper proposes a regularization term called MMA to promote angular diversity between pairwise weight vectors in DNNs. The method is easy to implement and improves performances in classification and recognition tasks. It can be easily applied in other tasks. 2. The paper gives detailed analyses of the gradient norm, thus the reason to use pairwise angles instead of cosine similarities is clearly explained. 3. The experiments demonstrate the effectiveness of MMA in both decorrelating weight vectors and improving network performances.

Weaknesses: The regularization coefficient is important. It can be seen from Figure 3 that the influences of different coefficients are unstable in a single experiment. To better clarify the influences, the author should report the average results of multiple experiments under the same setting. 2. In the Section 5, it seems that the proposed MMA can effectively improve the image classification performances. I wonder if the convergence speed and stability are changed after introducing the MMA regularization in the training process. Therefore, it would be better to give the training loss curves with and without MMA regularization. 3. The proposed method aims to decorrelating the network weights. Some works such as [1] [2] [3] that analyze and reduce the redundancy of convolutional filters are related to this direction. They should be discussed in Section 2. [1] Haase D, Amthor M. Rethinking Depthwise Separable Convolutions: How Intra-Kernel Correlations Lead to Improved MobileNets[C]//Proceedings of the IEEE/CVF Conference on Computer Vision and Pattern Recognition. 2020: 14600-14609. [2] Guo J, Li Y, Lin W, et al. Network decoupling: From regular to depthwise separable convolutions[J]. arXiv preprint arXiv:1808.05517, 2018. [3] Jaderberg M, Vedaldi A, Zisserman A. Speeding up convolutional neural networks with low rank expansions[J]. arXiv preprint arXiv:1405.3866, 2014.

Correctness: Yes

Clarity: Yes

Relation to Prior Work: Clearly

Reproducibility: Yes

Additional Feedback: The authors' response addressed my concern. So I maintain my rating (should be accepted)


Review 3

Summary and Contributions: ===UPDATE===== I have read the rebuttal as well other reviews. My concerns have been addressed so I keep my score. =============== The paper introduces maximal minimal angle regularizer (MMA) as a way to encourage the diversity of weight vectors in the layers of DNN. They analyze the optimization issues, suggest reasonable modifications, compare aganist baselines. Importantly proposed method and baselines are compared on the several Tammes problems (that is to place the number of points on a unit sphere in such locations that they maximize the distance between the closest neighbors) for which the optimal answers are known and it can be clearly seen that MMA outperforms analogues. Next the authors use MMA as a regularizer for DNN and show (1) improvements when the regularizer is added with optimal hyperparamter; (2) correlation between the minimal angle in DNN layers and test accuracy. Overall I think this method is of value for the community since it reveals that additional decorrelation really matters for improving generalization.

Strengths: The paper is methodologically solid with motivation, analysis of existing tools, careful comparison with baselines, ablation study and final experiments on a wide number of different architectures. The proposed method is computationally efficient and can be easily implemented into existing learning tools. I think it is relevant for the community.

Weaknesses: There are some questions/concerns however. 1. Haven't you tried to set hyperparameters for the baseline models via cross-validation (i.e. the same method you used for your own model)? Setting it to their default values (even taken from other papers) may have a risk of unfair comparison aganist yours. I do not think this is the case but I would recommend the authors to carry out the corresponding experiments. 2. It is unclear for me why the performance of DNN+MMA becomes worse than vanilla DNN when lambda becomes small? See fig.3-4. I would expect it will approach vanilla methods from above but from below.

Correctness: I think the claims and method are correct. The results seem to be convincing.

Clarity: The paper is clearly written and easy to follow.

Relation to Prior Work: The paper discusses different alternatives to encourage weight diversity. The closest analogues are described in more details and the proposed method is fairly compared aganist them.

Reproducibility: Yes

Additional Feedback: Compare aganist baselines whose hyperparamters were set via cross-validation


Review 4

Summary and Contributions: In this paper, authors propose a diversity regularization on filters in a neural network. There, normalized weight vectors of neurons are encouraged to be distributed uniformly on a hypersphere. Performance improvements are observed on cifar100 and TinyImageNet.

Strengths: + Removing strong correlations among filters seems to me a good idea. + Authors use the Tammes problem as the inspiration. + The analytical part seems sound.

Weaknesses: Authors may provide more insights and illustrations. For example, - On one side, I agree, removing filter correlation is a good idea. On the other side, more insights can be helpful to show uniformly distributed filters are actually the way to go. - Why does MMA show superior over orthogonal regularization in (14), which seems a very similar objective. - Orthogonality or uniform distribution are more often enforced over (inter-class) features. From this aspect, what is the direct effect on features using MMA? I notice some discussion around line 195, any additional illustration and supports? For example, in Table 6, will MMA alone match ArcFace performance?

Correctness: The method and empirical methodology look sound in general.

Clarity: The paper is clearly written.

Relation to Prior Work: The difference from previous contributions is clearly discussed.

Reproducibility: Yes

Additional Feedback:

[Author Response · NeurIPS 2020]

1 We thank the reviewers for their insightful comments. Below we respond to the comments point-by-point.

## 1 Response to Reviewer 1:

**The improvement achieved by MMA.** We appreciate the comments from the reviewer. Firstly, the difference of ours from previous researches is clearly discussed in Section 2. Secondly, we argue that the improvement over other methods is significant. For example, MMA regularization improves the TOP-1 accuracy of VGG19-BN by 1.65%, while the orthogonal regularization improves just by 0.75%. Besides, as a simple plug-in regularizer with negligible computational overhead, it is shown to be architecture-agnostic and produces consistent performance improvement on many tasks. Therefore, the key advantage of MMA regularization highlighted in this paper is not the significance but the robustness.

**Combined with other regularization methods.** Firstly, weight decay and data augmentation have been applied to all the classification models in our paper, as we exactly follow the same settings in the original papers proposing the models. Secondly, to further demonstrate this advantage, we perform a comparative experiment with the WideResNet-28-10 on CIFAR100: the accuracy of applying the AutoAugment [1] (SOTA data augmentation) is 82.65%, and combining the MMA regularization with AutoAugment gets a higher accuracy of 83.11%. Thirdly, the models of ArcFace and ArcFace+ have used weight decay and dropout. All these demonstrate that MMA can be combined with other regularization methods to further improve the test performance.

[1] Cubuk E D, Zoph B, Mane D, et al. Autoaugment: Learning augmentation strategies from data[C]//CVPR2019.

**Source of improvement.** Firstly, the claim that the lack of neuron diversity can hurt the generalization performance has been discussed in many previous work ([34, 35, 48, 20, 18, 31, 47, 19] of the paper's References), and we propose a simple and powerful method to solve this issue. Secondly, in all the comparative experiments, the hyperparameters and settings, including the random seed, are the same between models with and without MMA, so the comparisons are absolutely fair. Thirdly, as a plug-in regularizer without changing the architecture, the only effect of MMA is enhancing the neuron diversity. Therefore, we argue that the improvement of accuracy comes from increasing the neuron diversity.

**Training procedure.** Due to the limited space, we do not list the training details, but we illustrate that the settings follow the original papers proposing the models exactly. The batch size is set to 64 for DenseNet and 128 for other models. The learning rate is initially set to 0.1 and decayed by specific schedules. The optimizer is SGD with a momentum of 0.9. More details of the training procedure will be elaborated by a table in the final version.

## 2 Response to Reviewer 2:

**Regularization coefficient.** We appreciate the comments from the reviewer. Follow this suggestion, we report the average results of five runs with different random seeds in Figure 1 and Figure 2, and the influences become stable.

**Training curves.** Due to the limited space, we only supplement the training curves of VGG19-BN on

Figure 1: Coefficient tuning for VGG19-BN.    Figure 2: Coefficient tuning for ResNet20.

CIFAR100 as in Figure 3. The MMA gets persistently higher TOP-1 accuracy and lower loss than the baseline. Besides, the convergence speed and stability are not changed. We will plot more training curves in the final version.

**More related work.** We thank the reviewer for providing some related work. "Network Decoupling" and "Speeding Up" focus on accelerating the training and the evaluation of CNN respectively. "Rethinking Depthwise" introduces the BSConv based on intra-kernel correlations, while our MMA regularization targets at decreasing the inter-kernel correlations. We will discuss the details in the final version.

## 3 Response to Reviewer 3:

**Hyperparameters for baseline methods.** We appreciate the comments from the reviewer. We set the hyperparameters to the same as the papers proposing the methods, which conduct cross-validation experiments. To be more rigorous, we agree with the reviewer and will implement cross-validation experiments by ourself in the final version.

Figure 3: Training curves of the VGG19-BN.

**The cases of small lambda.** As the reviewer 2 stated, this is because the influences of different coefficients are unstable in a single experiment. To better clarify the influences, we report the average results of five runs with different random seeds in Figure 1 and Figure 2, in which the performance of small lambda approaches vanilla methods from above.

## 4 Response to Reviewer 4:

**More insights.** We appreciate the comments from the reviewer. Firstly, based on the Tammes problem, the uniform distribution means the minimal angle is maximized, therefore the angular diversity is boosted to the utmost extent. Secondly, many previous researches ([34, 38, 20] of the paper's References) demonstrate that large angular diversity can decrease filter correlation. Therefore, we argue that uniformly distributed filters are the way to remove filter correlation.

**Comparison of MMA and orthogonal regularization.** Firstly, MMA focuses on the uniform distribution, while orthogonal regularization pursues orthogonality for all the pairwise vectors. Secondly, the gradient analysis of orthogonal regularization is similar to the one of cosine loss in Eq. (5), whose gradient is not as stable as the MMA. Thirdly, Figure 1 of the supplementary material demonstrates that MMA gets the largest minimal pairwise angle across all layers, and therefore the most diverse weight vectors. Fourthly, as discussed in [20] of the paper's References, orthogonal regularization tends to group neurons closer, especially when the number of neurons is greater than the dimension.

**The direct effect on features.** For output layer, MMA aims to maximize the distances between classifier neurons. Each classifier neuron plays the role of ground truth centre of features falling into the corresponding category. Therefore, MMA regularization can maximize the inter-class feature separability. ArcFace can enhance the intra-class compactness explicitly and inter-class discrepancy implicitly, while MMA regularization only focuses on the inter-class separability. So, it is not fair to compare MMA alone with ArcFace. However, the MMA regularization can further boost the accuracy of ArcFace and NormFace. We will investigate the effects of MMA on other feature learning methods and tasks.

[Meta-Review · NeurIPS 2020]

The paper has initially received mixed reviews, but post-rebuttal the expert reviewers have converged to the decision that the paper is above the acceptance threshold and that the proposed regularization is of wide interest to the community. Accept. The authors are encouraged to incorporate the extra experimental results from the rebuttal into the final version of the paper. Also, the related work section should be revised by incorporating relevant works pointed by the reviewers (and as promised in the rebuttal).